# The Power of Place: Unleashing the Potential of Place-Based Green Energy Landscapes

William Glockner *  , Krista Planinac and Kirk Dimond

College of Architecture, Planning and Landscape Architecture, University of Arizona, Tucson, AZ 85721, USA; kplaninac@arizona.edu (K.P.); kirkd@arizona.edu (K.D.)
* Correspondence: wglock01@arizona.edu

**Abstract:** This research explores the role landscape architects can play in shaping renewable energy infrastructure in the Southwest United States. Conventional energy development often neglects the impacts on landscapes and communities, resulting in community frustration and project terminations. To address this issue and tackle the need for decarbonization, the Southwest Regional Virtual Workshop was convened to foster co-creation and generate innovative ideas for new energy solutions. The Southwest Regional Virtual Workshop (SRVW) aimed to unite landscape architects, architects, engineers, and energy professionals to craft place-based, at-scale, and environmentally sensitive solutions. Key insights from this study demonstrate landscape architects have the capacity to help transform renewable energy projects into attractive, engaging, and productive infrastructure. Their expertise in community engagement, site-specific design, and interdisciplinary collaboration positions them as ideal designers for energy landscapes that go beyond mere functionality. By adopting a landscape-centric approach, landscape architects can help seamlessly integrate energy infrastructure with the environment and aesthetics to gain steadfast community support. Harmonizing functionality with visual appeal can instill a deep sense of pride and ownership among community members, ultimately fostering increased acceptance of renewable energy development. In conclusion, landscape architects can expand upon their expertise to include energy and help create projects that align with the values of local communities and contribute to a resilient energy future.

**Keywords:** landscape architecture; renewable energy; southwest; decarbonization; community acceptance; aesthetics; land use; sustainability

## 1. Introduction

The Southwest is a region where the interplay between solar and hydrological factors has shaped the physical, biological, social, cultural, and political aspects of the region [1–4]. Although the Southwest United States is a commonly used term, one may be surprised to learn that there is no official agreement on the regions it encompasses [5,6]. For the purposes of this research, experimenters use the term 'Southwest' to represent the states of Arizona, California, Nevada, New Mexico, Texas, and Utah. The physical landscapes of the Southwest range from snowcapped peaks to low-lying deserts, rivers, coasts, and dramatic canyons [6]. Parched and barren earth contrasts with fertile flood plains nourished by expansive watersheds, groundwater, and seasonal monsoons [7]. Unfortunately, climate change exacerbates these extremes, bringing record heat, forest fires, diminishing snowcaps, and straining life-supporting water supplies [8–11].

Despite these challenges, the Southwest's landscapes still foster biodiversity, with unexpected flora and fauna adaptations for survival [12–14]. Indigenous peoples have lived and worked the land for millennia, and contemporary habitation relies on substantial energy inputs and resource extraction [2,4,15–18]. Examples of substantial energy generation and resource extraction can be seen in many of the Southwest's iconic energy landscapes, such as the Hoover and Glen Canyon dams. Both sites forever altered the land they were

constructed on, centralized power generation, and impacted miles of wilderness to support extensive networks of power lines and aqueducts [19–21].

It is noteworthy that Hoover and Glen Canyon dams, two of the most dramatic instances of twentieth-century energy landscapes, had no dedicated landscape architecture firm involved [22,23]. The Hoover Dam was commissioned by the United States Bureau of Reclamation (USBR) and was primarily the work of The Six Companies, with architectural credit to Gordon B. Kaufmann [19–21]. Notably, a smaller design professional on the build, sculpture Allen Tupper True, chose to go beyond the Greek, Roman, and Egyptian motifs and opted to include place-based elements, such as Navajo artwork, in the dam's final design [24]. In the instance of Glen Canyon, the project's conception was commissioned and overseen by several government organizations: the United States Geological Survey (USGS), the USBR, and the Army Core of Engineers [22,23]. Along with these agencies, the main private builder of the dam was Merritt-Chapman & Scott Corporation, although there was notable involvement from the Southern California-Edison Company and the Phoenix Cement Company [22,23,25]. Although these projects re-defined the Southwest's energy landscape and are undeniable triumphs of engineering, both projects would struggle to be replicated today due to concerns around the enormous financial, political, and environmental cost of such developments [26,27].

These projects showcase how key twentieth-century energy developments, led by government agencies and private institutions, have prioritized engineering marvels and extractive energy projects, while overlooking environmental and community impacts [26,28]. Legacy development has often viewed the landscape more as a commodity than a collaborator in design [28]. In transitioning to renewable energy, we must acknowledge past shortcomings and adopt an improved approach for a sustainable future [26,28]. New forms of energy landscapes will have to focus on a series of smaller interventions that enhance the region they are built in and are able to be implemented quickly at scale to meet our growing power needs while garnering community support [29,30].

One framework for the next generation of renewable energy landscapes can be found in the methodology of creating place-based, at-scale infrastructure. Place-based at-scale (PBAS) represents the goal of aligning a technology or infrastructure with community goals in a manner that promotes replicability and scalable solutions nationwide, thereby hastening the transition towards cleaner energy systems without sacrificing the uniqueness of the communities these landscapes serve [31]. PBAS designs are place-based in the sense that they cater to and work with each location where they are installed, and their modularity does not sacrifice or override the uniqueness of any given installation. PBAS technologies can be likened to the way phone cases all provide the same primary function of protecting a device, but they do so in a scalable and unique fashion. Individuals can choose a phone case that best represents both their unique needs—such as wallet storage, easy grip, or waterproofing—alongside choosing a design that reflects their personality and interests. Bringing PBAS energy landscapes to Southwestern communities will enable each community to receive an energy solution that fits their needs, and aids in the universal shift towards expanding green energy while avoiding community opposition.

PBAS energy landscapes have the potential to garner community support more effectively by accounting for the community's diverse needs, values, and culture [31]. An example of this can be seen in the solar photovoltaic system installed at Burbank Power and Water in Southern California (Figure 1). The photovoltaic system advantageously provides shade to employee vehicles while capturing the plentiful sun rays in this climate. Meanwhile, panels are intentionally angled in a way to reference the aviation history of the community. Furthermore, the structural support that elevates the panels doubles as part of aesthetic pillars and fencing that matches the Art Deco architectural style of the historic campus and even includes rain chains for stormwater management. The site seamlessly integrates with a community-facing "green street" along a previously neglected boulevard. Through this approach, the site not only tackles climate challenges, drought resilience, and energy requirements but also celebrates the community's rich and distinctive heritage.

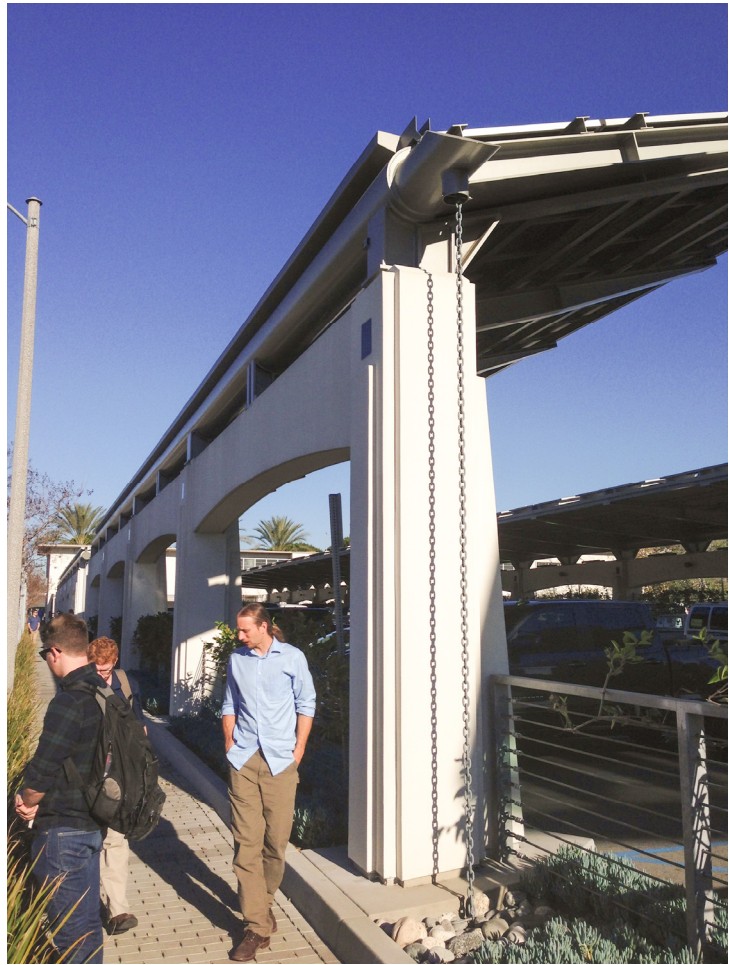

**Figure 1.** Solar Photovoltaic covered parking at Burbank Water and Power exhibits place-based integration in architectural detail and incorporation with stormwater management (Photo credit: Kirk Dimond).

O'Neil et al. emphasized six pathways toward PBAS energy landscapes to include (1) meaningful multifunctionality, (2) respecting and leveraging natural capital, (3) generating local value, (4) decentralizing energy generation at a local scale, (5) ensuring resilience against climate disruptions, and (6) promoting energy justice [31,32]. Neglecting these pathways may result in a community deeming renewable development as not worthwhile and impeding the rollout of green energy. Instances of recent community resistance towards new, renewable energy developments can take the form of moratoriums on solar and wind energy development or creating difficult zoning requirements and challenging setbacks [33]. By recognizing the multifaceted nature of these communities and working with them instead of against them, it will be possible to roll out new PBAS energy solutions more quickly.

In addition to community support, strategic public sector policies and actions can influence long-term planning horizons and broad forms of innovation to enable technical and social advancements [31]. Recently, the public sector has introduced several key pieces of legislation that encourage the creation of PBAS energy landscapes [31]. A recent $1.2 trillion infrastructure bill will provide direct investments in the electric grid in addition to funding for a range of renewable energy projects [34]. In addition to new funding, the government has set a target to reduce greenhouse gas pollution from 2005 levels by 50% by 2030 [35]. The urgency and scope of this target emphasize the need to design and deploy energy technologies that scale well. If energy landscapes face scalability issues or encounter significant community resistance, meeting these targets will become tenuous. Setting a

key goal and allocating funding highlights the public sector's key role in advancing green energy landscapes and illustrates that the push towards a renewable energy future is not solely the purview of the private sector.

The act of problem solving around renewable energy typologies, appropriate to the place, is increasingly critical as we tackle the grand challenge of decarbonization. Many past designs from the 20th century have prioritized maximizing energy and economic outputs at unprecedented scale and across sensitive landscapes, resulting in many negative environmental, social, and economic impacts [36–39]. Traditional energy development has driven significant imbalances between energy companies, communities, urbanization, and the natural landscape [40]. The beginnings of our renewable energy transition are underway and are driving rapid changes to land use at concerning rates [41]. The rapid transformation of our grid is currently hallmarked by community resistance and conflicts that impede the progress of green energy implementation [42]. Despite being a landscape and community challenge, renewable energy development teams have long lacked involvement from landscape architects and other design professionals that holistically consider community and environmental factors in place-based development [43]. This exclusion may be an oversight, as sustainable energy landscapes represent a space where landscape architecture and other fields would be best served to work together to achieve global sustainability goals, empower local communities, and protect landscape quality [43].

To investigate this opportunity for landscape architects to contribute to the rollout of PBAS renewable energy, researchers formed the SRVW to generate co-created ideas for at-scale renewable energy solutions. The SRVW strives to foster the co-creation of strategies and principles that will be shared with the Pacific Northwest National Laboratory (PNNL), the U.S. Department of Energy (DOE), and other energy professionals to help move toward fostering a new energy perspective for our region and nation. A perspective where renewable energy development optimization is PBAS with enhanced sensitivity and understanding of the impacts on the diverse landscapes where we live, work, and play. The core mission of the SRVW is to collaboratively co-create ideas, strategies, and principles with stakeholders that advance a new PBAS energy perspective for the region. Although not its primary goal, the SRVW also hopes to foster connections between design professionals and energy professionals in a virtual environment that encourages collaborative participation. The SRVW will have been a success if it provides an avenue for collaborative, co-created ideas, strategies, and principles on PBAS energy infrastructure that help shape future energy efforts in our region and nation. The main pre-requisite for prospective SRVW participants was a willingness to hear and share ideas in a transdisciplinary environment and collaboratively generate place-based approaches to advance the future of renewable energy development.

Renewable energy already offers many communities immense benefits and will likely serve to power much of our economy in the decades to come [37,41,44]. So, the value and desire for a successful transition towards renewable energy are vitally important. Numerous research papers have showcased the power of green energy and highlighted the collective need to begin switching to more renewable and less impactful forms of energy for humanity to continue to thrive and live in greater harmony with the planet [36–39,45–48].

In the face of this opportunity, researchers explored the potential role of landscape architects, focusing on several key articles and studies that underscore their current role and potential future contributions. The literature reviewed helped create a framework suggesting that landscape architects could be a key contributor to balancing community and commercial desires in the renewable energy sector through effective design and planning [43,44,49].

For instance, Dean Apostol's article, "The Renewable Energy Landscape: Preserving Scenic Values in our Sustainable Future", delves into the importance of safeguarding scenic values in renewable energy project development. While renewable energy is vital for sustainability, it can also pose visual landscape challenges. Apostol suggests strategic measures to preserve scenic values, including the placement of wind turbines and integrating

landscaping to harmonize projects with their surroundings. Community involvement in the planning process is emphasized to address local values and concerns [36,50].

Pasqualetti's article highlights the social barriers that can impede renewable energy landscape development. Social and cultural factors, institutional resistance, and regulatory processes all influence the adoption of renewable energy technologies. The aesthetic quality of landscapes can significantly impact public perceptions. Engaging with communities and stakeholders is crucial for designing visually and socially acceptable projects. Streamlining regulatory processes can facilitate renewable energy development [51].

Musall and Kuik's study, in Germany, examines the local acceptance of renewable energy projects. It reveals that economic benefits, environmental concerns, and perceptions of landscape impact public acceptance [47]. Community participation in decision-making processes is essential, with landscape architects playing a vital role in integrating local values into project design. For instance, "Renewable energy policy and public perceptions of renewable energy: A cultural theory approach" by West, Bailey, and Winter suggests that the influence of visual impacts and cultural biases on public perceptions of renewable energy projects is vital for project completion [42]. This understanding is essential for designing effective policies and improving communication, both of which are necessary to mitigate the negative effects associated with renewable energy projects.

Azarova et al.'s study explores the design of local renewable energy communities to enhance social acceptance. The survey conducted in several countries reveals a preference for smaller-scale, community-led projects. Tailoring projects to local preferences can help foster social acceptance [38].

The literature reviewed showcases that landscape architects possess the ability to help civic energy projects receive community endorsement and get through critical development stages. Therefore, it is important to consider how landscape architects may best shape the future of energy generation. The article "Power Player", as featured in Landscape Architecture Magazine, highlights how following the six design principles can help contribute to the green energy transition in a fashion that prioritizes multifunctionality, preservation of natural capital, generation of local value, encourages decentralization, mitigates climate risks, and promotes energy justice [44]. These six principles would form the main points of investigation for the Southwestern Conference and helped guide the co-creation session around the topic of multifunctional land use, respecting scenic qualities, community input, ecological impact considerations, and ensuring communities benefit from all energy landscape development.

The six principles, the PBAS design methodology, the history of energy landscape development, and the connection between landscape architecture and community endorsement were distilled by the authors into several key topics. The following topics listed helped guide participants in the SRVW as they collaboratively co-created ideas, strategies, and principles with stakeholders to advance a new energy perspective for the region:

- How can the six pathways of PBAS energy landscapes best be incorporated into a project?
- What natural opportunities and trade-offs exist between the six pathways?
- What is the most effective form of PBAS energy generation?
- What key land use issues routinely derail PBAS energy projects?
- What is the role of landscape architects in the future of green energy?
- Are there any novel opportunities to capitalize on existing infrastructure?

The SRVW hoped to answer these key questions, improve upon current understanding, further investigate the six pathways, and, ultimately, brainstorm, discuss, and develop novel forms of PBAS energy landscapes that can be more easily integrated into the American Southwest. By bringing together experts from various disciplines and backgrounds, the conference seeks to facilitate brainstorming, discussion, and the development of innovative approaches for integrating PBAS green energy into the American Southwest's energy landscape. The conference will provide a platform to explore how multifunctionality, natural capital, local value, decentralization, climate risks, and energy justice can be effectively addressed and integrated into future PBAS energy systems. By engaging in

co-creation across disciplines, researchers hope to better understand the role that landscape architects can play in helping the rollout of renewable energy research and practice. The virtual conference took place over the span of a day and featured several targeted breakout sessions followed by a period of design synthesis and critique. Since the event was a virtual conference, most ideas were shared with text descriptions, photo uploads, basic sketches, and linked articles.

The principal findings from investigating these questions are that landscape architects are well positioned, as a profession, to rectify some of the shortcomings, and unsustainability in legacy energy development and help shape the future of PBAS renewable energy landscapes while balancing environmental, social, and economic concerns.

## 2. Materials and Methods

To facilitate innovative approaches for the transition to renewable energy in the Southwest United States, the SRVW adopted a co-creation methodology. Co-creation, a collaborative process involving diverse participants, was employed to generate novel concepts and solutions with a focus on sustainability and stakeholder engagement. The University of Arizona hosted the SRVW on 9 January 2023, with sponsorship from the PNNL and the DOE. The primary aim of the SRVW was to develop fresh perspectives and principles concerning PBAS renewable energy infrastructure.

To promote awareness and attract the target audience, a multi-faceted marketing strategy was devised. This strategy included leveraging tools such as the American Society of Landscape Architect's (ASLA) Firm Finder, connecting with ASLA chapters in the region, researching Southwest-based landscape architecture projects, and fostering relationships with organizations like The National Association of Minority Landscape Architects and the Council of Educators Landscape Architecture (CELA). A contact database of over 100 professionals within the Southwest United States was assembled, and invitations were extended via email and social media. Upon registration, attendees received confirmation emails, and a reminder email was dispatched five days prior to the SRVW.

Concurrently, a student ideas competition, sponsored by CELA, was conducted. Open to graduate and undergraduate students from CELA Member Schools in the Southwest region, this competition called for participants to submit drawings and explanations of their renewable energy ideas. The Student Design Competition presented an exciting opportunity for participants to envision a layout for a Southwest renewable energy PBAS landscape. Researchers crafted a flyer and distributed it through CELA members. To participate, students simply followed a QR code or the link on the flyer and adhered to the design brief.

Participants were required to choose a Pathway aligning with their vision from six available options: Multifunctionality, Natural Capital, Generating Local Value, Decentralization, Resilience to Climate Disruption, and Energy Justice. Detailed information about each pathway was provided through a link to a past paper by the authors. Students were then tasked with expressing their ideas through a drawing, specifically focusing on the Southwest United States. There were no restrictions on the scale of the landscape or chosen pathway, and students were provided with the flexibility to use digital and/or analog methods. After receiving the submissions, a multi-disciplinary team of university professors in the fields of Architecture, Landscape Architecture, Urban Planning, and Civic Design evaluated the student projects anonymously. Ratings were based on relevance to the prompt, feasibility, novelty, artistic quality, and potential for installation in a PBAS fashion.

This opportunity allowed students to contribute innovative ideas to the future of renewable energy landscapes in the Southwest. The competition aimed to reward excellence, foster student work, generate new ideas, and provide winners with an opportunity to present their concepts at the SRVW workshop, fostering creativity and co-creation. The winning designs served as a catalyst for discussions during the conference, and several examples of student work have been featured below (Figure 2).

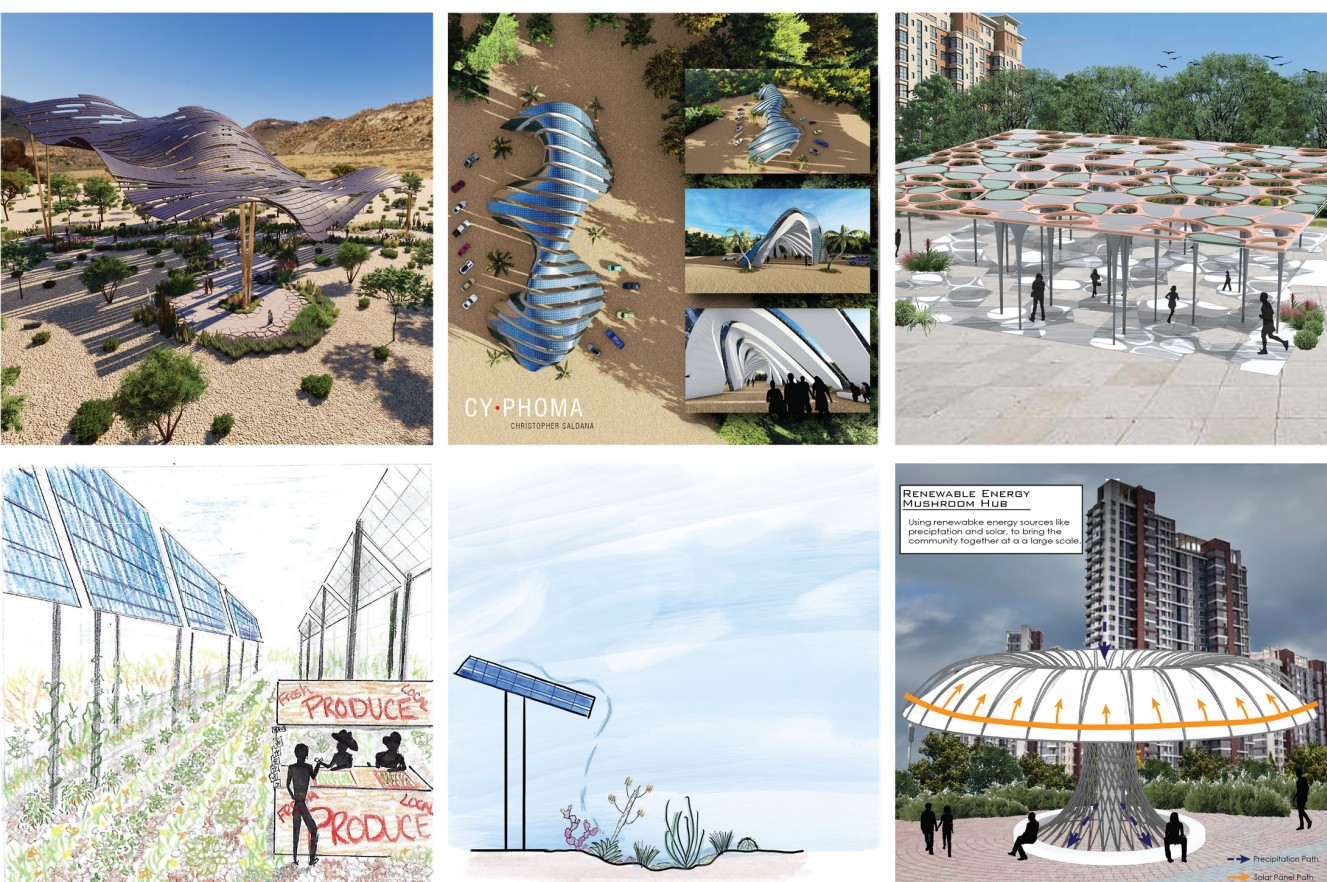

**Figure 2.** Samples of student design solutions both rendered and drawn. Student names in order of work depicted left to right, top to bottom: Brendan Berry, Christopher Saldana, Alyssa Gainey, Jessica Eppard, Oscar Rodriguez Ponce, Cloudia Wooten.

The SRVW followed a structured co-creation process to foster idea sharing among participants with diverse backgrounds. The process was centered on three primary contexts in the Southwest United States: low desert, chaparral, and high desert/plains. The SRVW schedule was outlined as follows:

1.  A welcome and introduction session was conducted by the University of Arizona, PNNL, and the Department of Energy.
2.  An overview presentation of the six pathways designed for the transition to renewable energy.
3.  A segment was dedicated to the student ideas competition, during which winners presented their award-winning designs.
4.  A tutorial on how to utilize Conceptboard, the chosen virtual collaboration tool.
5.  Phase I: Participants engaged in a brainstorming session within breakout rooms, each focusing on one of the six pathways.
6.  A lunch break during which moderators analyzed common themes from the pathway sessions.
7.  Phase II: Participants explored synergies and tradeoffs between paired pathways.
8.  Phase III: Each paired group presented principles and key ideas, followed by discussions and prioritization.

The structure of the SRVW was carefully designed to encourage active collaboration, idea generation, and the co-creation of innovative principles and solutions for renewable energy infrastructure in the Southwest United States. For more on the structure of the conference, please see Figure 3 below.

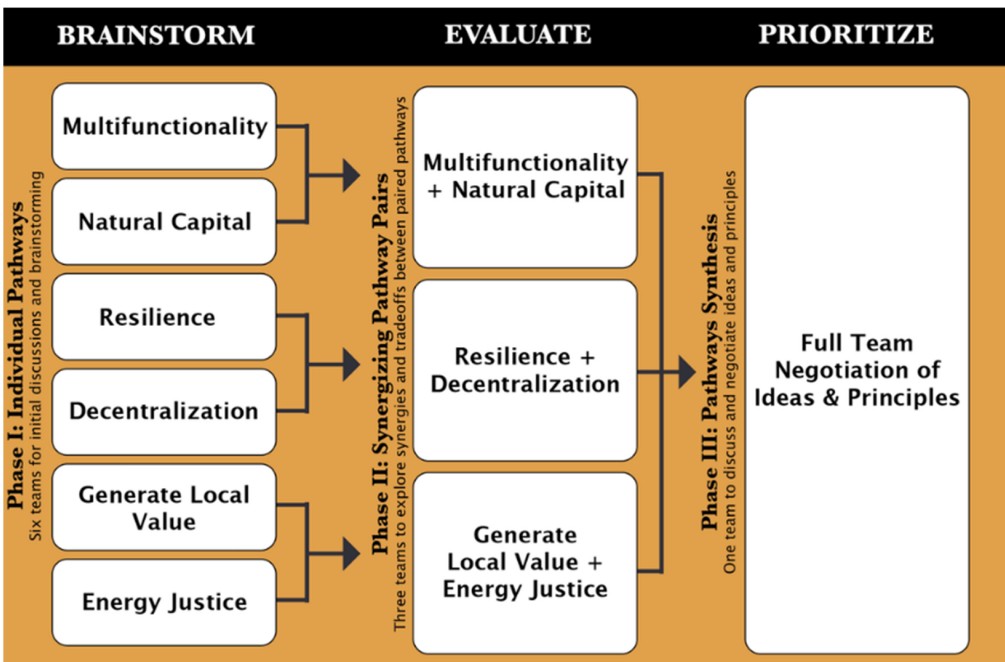

**Figure 3.** Co-creation process diagram with three phases of group participation.

As for key demographic and participant information on the workshop, the attendees enrolled in the workshop via Zoom.us, as the conference was entirely virtual, and the attendees were prompted to provide information about their state of residency, profession/field/industry, and their primary area of interest. While a total of 93 individuals registered for the workshop, their active participation at any given time ranged between 35 and 40, with some participants joining and leaving at different points during the workshop. The collected responses from all registered participants are outlined in the table below (Table 1).

**Table 1.** Table showing group composition by region, profession, and primary pathway interest.

| State | Number | Percent |
|---|---|---|
| Arizona | 42 | 45% |
| California | 22 | 24% |
| Utah | 7 | 8% |
| Nevada | 4 | 4% |
| New Mexico | 4 | 4% |
| Texas | 4 | 4% |
| Other/No Response | 10 | 11% |
| **Profession/Field** | **Number** | **Percent** |
| Landscape Architecture | 49 | 53% |
| Architecture | 18 | 19% |
| Sustainability | 7 | 8% |
| Engineering | 4 | 4% |
| Other/No Response | 12 | 13% |
| **Pathway of Interest** | **Title 2** | **Title 3** |
| Multifunctionality | 30 | 32% |
| Climate Resilience | 27 | 29% |
| Generate Local Value | 17 | 18% |
| Energy Justice | 11 | 12% |
| Decentralization | 9 | 10% |
| Natural Capital | 3 | 3% |

## 3. Results and Discussion

In this section, researchers present the key participant findings derived from a comprehensive review and analysis of the discussions and ideas generated during the SRVW. The SRVW involved a diverse group of participants who engaged in detailed discussions across six distinct pathways for several hours. The conference generated a great deal of ideas and highlighted numerous potential opportunities. These findings are instrumental in understanding the future development of PBAS energy landscapes in the Southwest region. A few samples of the work conducted on Conceptboard at the SRVW have been presented below in Figures 4 and 5.

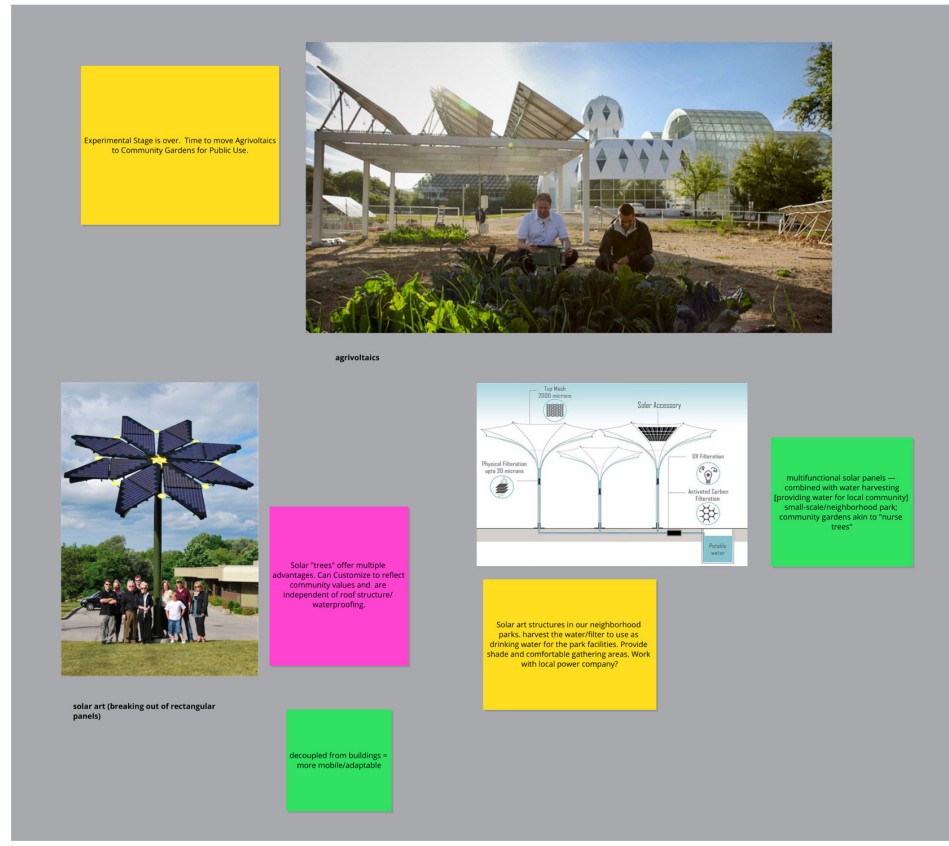

**Figure 4.** A small clip from the SRVW concept board showcasing participant ideas and collaboration.

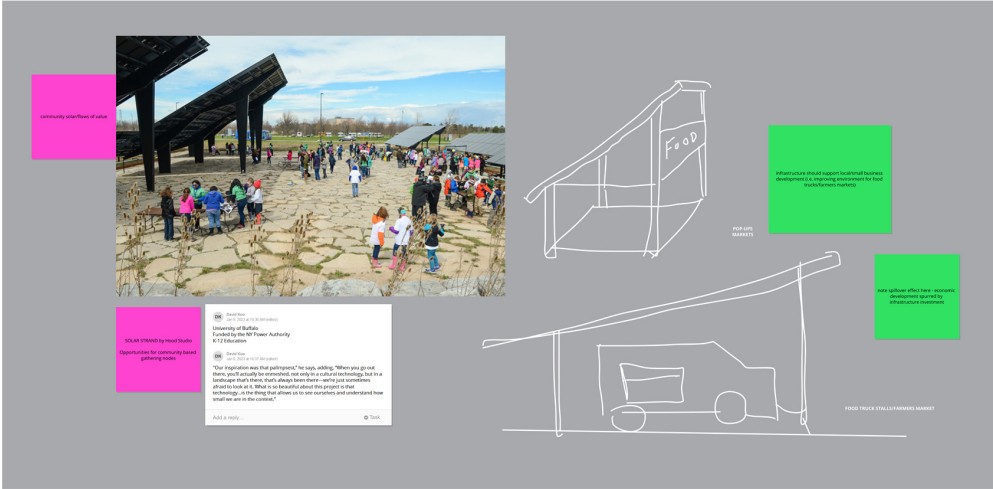

**Figure 5.** Another small clip from the SRVW concept board showcasing participant collaboration and sketched ideas.

### 3.1. Individual Pathway Findings

The first group, the multifunctionality group, discussed the challenges of achieving multifunctionality in renewable energy infrastructure design. Many members of the group prioritized other benefits, such as aesthetics and community buy-in, over optimizing energy production itself.

The second group, the natural capital pathway group, emphasized the importance of protecting nature when constructing renewable energy projects in rural areas. The group proposed educational campaigns and research to optimize the design of renewable energy projects that allow for the restoration of a dynamic ecosystem.

The third group, the resilience group, focused on assessing current energy consumption and ways to reduce the impact of the built environment through passive design and the use of native landscape.

The fourth group, the decentralization group, identified solar energy as an ideal technology for decentralized systems due to its high efficiency and cost-effectiveness. Decentralized, community-owned micro-grids powered by solar energy were seen as a way to significantly lower energy costs for residents and provide greater protection from power outages.

The fifth group, the generate local value pathway, discussed the economic development opportunities of renewable energy systems but also expressed skepticism and concerns about the manufacturing and installation costs of solar systems. The participants felt that creating electric microgrids locally or regionally was one way to increase power production while generating local value.

The sixth group, the energy justice group, discussed the importance of an equitable distribution of the benefits and costs of renewable energy systems, with an emphasis on considering the needs and perspectives of marginalized and low-income communities. The group concluded that communities should have a strong voice in decisions made regarding renewable energy projects.

### 3.2. Synergystic Overlaps between Pathways

Following the six group discussions, researchers analyzed the topics that emerged and identified key themes and ideas that would be used in the next phase of the SRVW. The researchers grouped participants' ideas and conclusions, identifying overlaps, synergies, and unique aspects for all teams and an overview of these synergies and trade-offs can be found in Figure 6. Researchers used a Venn diagram model to quickly assess several trends and synthesize the findings.

The Multifunctionality group and the Natural Capital group had clear overlaps in the themes across both pathways. Both groups emphasized the importance of short distances between energy production regions and energy consumption sites. Participants also discussed the importance of pursuing new potential technologies, such as "PV paint" or other novel photovoltaic materials that could support synergies. However, the specifics of these new technologies and the data underlying their effectiveness were not well-known by most of the participants and, potentially, do not exist or function in the fashion that participants had envisioned.

The second group, Decentralization and Resilience, discussed the synergies and trade-offs to uncover novel ideas for the future of secure energy generation landscapes in America. The key synergy participants identified was how decentralized community microgrids were inherently more resilient than centralized energy generation sites. Community microgrids could promote local energy production, reduce reliance on the grid, and foster a sense of shared ownership and responsibility. Another synergy identified was that decentralized energy systems could reduce the impact of natural disasters, particularly in fire-prone communities, by eliminating the need for excessive and large transmission corridors that are known to be potential sources of ignition for wildfires.

| | SYNERGIES | TRADE-OFFS |
|---|---|---|
| **Multi-functionality & Natural Capital** | • Benefit from enhanced policy and regulation based change<br>• Benefit from habitat corridors along parking lots and medians<br>• Offer immense opportunities for shade structures and preservation of trees<br>• Effective land management helps reduce the impact of new systems<br>• Can be integrated into a sense of subjective aesthetics in civic centers if natural aesthetics are desired | • Breaking up panel installation to integrate trees may be more costly to maintain and less energy productive<br>• If you build a multi-functional site, you are still disturbing the environment<br>• Solar panels and other energy generation shade structures are not as pretty or natural as trees<br>• Multi-functional development could result in disturbing regions that are currently being already used putting additional stress on the local ecosystem |
| **Resilience & Decentralization** | • Benefit from the removal of national, governmental red tape and a greater empowerment of local communities<br>• Empower communities to respond to disasters more quickly<br>• Decrease the odds of total community blackouts and power outages<br>• Can be leveraged to systems outside of energy generation that quickly empower communities to chart their own destiny<br>• Mitigate systemic biases in the political and economic sector by empowering people directly | • Most desired scale of power generation<br>• Not compatible with government interest and establishment conventions<br>• How locally generated energy is managed, stored, and used<br>• Decentralization can decrease resilience if a decentralized system is pursued to a far enough extent that it is removed entirely from a main grid |
| **Local Value & Energy Justice** | • Benefit from being smaller scale<br>• Allow for equal protection for all in a community<br>• Encourage communities to determine what is most important to them and engage in self governance<br>• Providing power from a source directly to the end user can help with cost but may require an outside party to own the system to control overall energy flow<br>• Community perceptions towards energy and renewable can expedite acceptance<br>• Opportunities for micro-finance and local and decentralized banking and money lending | • New hydro dams can displace residents and hurt local communities by not considering them in the decision process<br>• How local are benefits concentrated. Who owns the output of community based energy systems.<br>• Avoid furthering local discrepancies in living conditions and economic mobility<br>• Community perceptions toward development and renewable energy can hinder development in key regions<br>• To what extent to experts and expertise limit a communities ability to engage with green energy |

**Figure 6.** Overview of synergies and trade-offs for SRVW.

The third group, Generate Local Value and Energy Justice, identified the need for a different energy generation approach, one that is balanced and focused on the needs of the community. The idea of integrated design was discussed, as was the idea of having landscape architects in decision-making roles at the community level. Here, too, the conversation turned to community and what various scenarios could look like if the development of renewable energy systems prioritized those who lived within a region. The primary concern of participants was how to ensure that local residents were on the receiving end of the social, cultural, and economic benefits of new energy infrastructure. Participants identified that the ability to capitalize on local talent would bring economic opportunities to the community. Specifically, prioritizing the use of local talent in operating, maintaining, and building PBAS energy generation landscapes would generate more distributed work opportunities and high-wage opportunities.

After these joint sessions, one speaker was nominated from each group to summarize their findings and provide insights into their group's discussion process. The collected findings from each group were then compiled and evaluated against an Eisenhower matrix, a decision-making tool that measures the level of impact a project might have versus the amount of effort it would take to implement. This matrix allowed for a structured and systematic assessment of the potential impact of the various concepts discussed during the session and a synopsis of that matrix is shown below (Figure 7). Through this evaluation process, researchers were able to prioritize the SRVW's findings and determine which ideas held the most promise for future research and implementation.

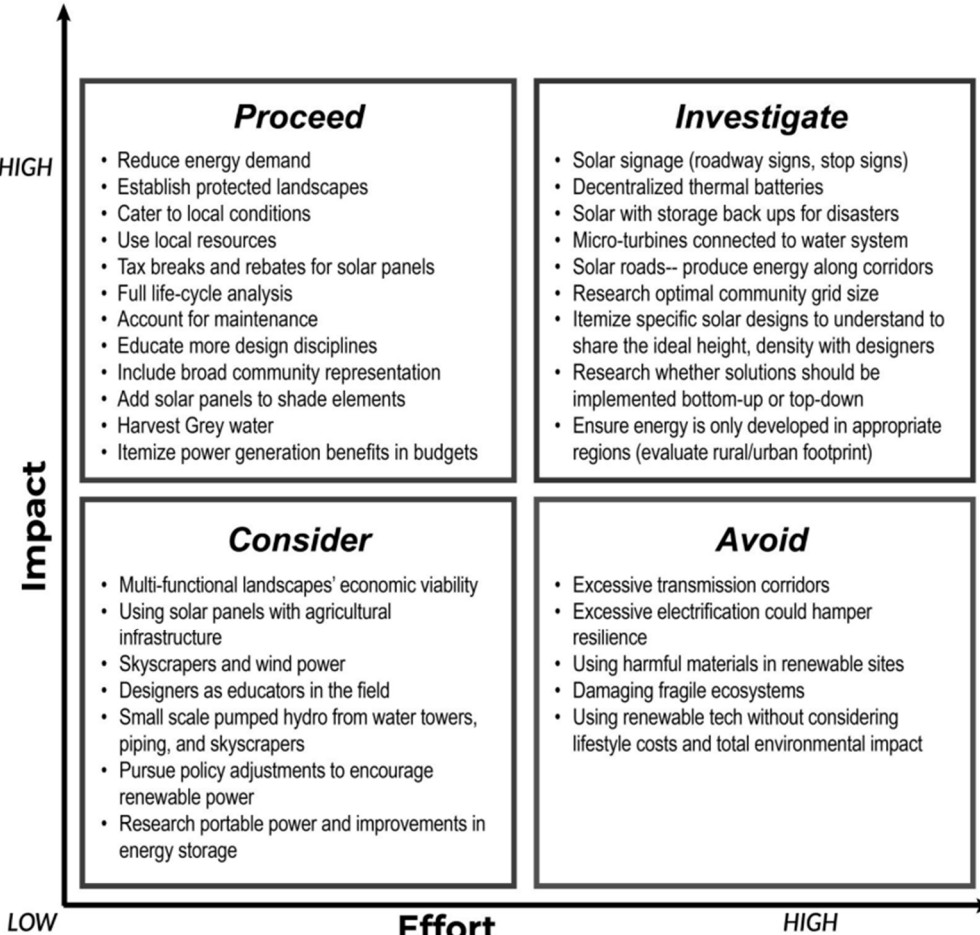

**Figure 7.** Southwest Workshop ideas and principles as ranked and categorized by participants in an Eisenhower matrix.

*3.3. SRVW Methods Review and Analysis*

The participant-generated ideas underscored the importance of small, decentralized energy landscapes designed by landscape architects to address the urgent need for green energy adoption. Participants found that incorporating PBAS environments into renewable energy production can benefit local communities while preserving existing wildlife and reducing climate vulnerabilities. The SRVW also found that participants were willing to prioritize aesthetics and community buy-in over energy production optimization, emphasizing the importance of community involvement in designing and taking ownership of their energy infrastructure. Participants also noted that empowering communities through PBAS renewable energy landscapes can rapidly expand the reach of green energy technologies.

The conference had many wonderful suggestions and ideas. General consensus amongst researchers was that the following co-created ideas: avoiding developing energy infrastructure in undisturbed lands, reducing evaporation from reservoirs and canals with agrivoltaics/flotovoltaics, civic pilot projects, green land art, and microgrids were promising solutions for decentralized energy production and avoiding transmission issues. Microgrids and decentralized energy redundancies were identified as also being in the interest of disaster relief efforts and national security. The conference homed in on the indelible link between water and energy, with many participants highlighting the potential of micro-hydro power as a solution and generating ideas for multi-functional, community-owned sites. The discussions around these multi-functional sites underscored the valuable role that landscape architects currently play in the realm of stormwater reclamation at a community level and shed light on the role landscape architects could play in the future of sustainable energy sites. There was also a consistent cry for policymakers to shift policies to promote the development of renewable energy in already developed regions to make progress towards a sustainable energy future; however, these concerns with policy and the political process remained largely amorphous for the duration of the conference. As such, no discrete, actionable political solutions were proposed by participants, and even if complete and detailed policy interventions had been proposed, they would be outside of the scope of our research.

Throughout the SRVW, participants emphasized the critical role that landscape architects can play in helping with the creation of small, decentralized energy landscapes that benefit both local communities and the environment. However, landscape architects alone cannot achieve this transformative energy revolution. A multi-disciplinary approach is needed, involving interdisciplinary teams that balance technical expertise with creativity and resources to develop innovative energy landscapes. Future workshops and project development phases should include more engineers, architects, policy makers, and other design professionals to provide a more holistic perspective on proposed solutions and assess their feasibility. Overall, the conference was successful in co-creating numerous creative energy solutions, confirming that PBAS renewable energy landscapes offer a promising new direction towards a sustainable future, and showcasing that landscape architects have a unique opportunity to help create targeted and effective solutions that empower communities and benefit the planet.

The SRVW's findings indicate that better alignment across disciplines is necessary to support PBAS renewable energy in the Southwest. Designers understood the conceptual challenges of energy landscapes, but knowledge gaps limited greater creativity beyond popular energy solutions such as tidal, solar, wind and vertical layering. As a result of this knowledge gap, many participants' suggestions seem promising but may not be feasible in the exact state they were pitched and, consequently, will require further investigation to fully implement. For instance, participants' suggestions around community-owned and operated micro-grids will require a wealth of cooperation between various stakeholders to determine fair and feasible co-ownership, even if they represent a very promising development for the future of decentralized green energy [52]. Another of the solutions pitched, micro-hydro turbines implemented on existing city water infrastructure, would require coordination between various agencies and experts to ensure that they are designed

and implemented effectively. Moreover, this technology would have to be evaluated by many professionals to ensure that the cost of such an intervention would yield a positive, cost-effective energy output from the project. Many of the cutting-edge technologies that were proposed and discussed at the conference by landscape architects remained somewhat speculative. Although the conference was well suited to design these novel sites in a fashion that includes community desires and functions in a beautiful space, some of the technological findings may need further investigation to determine feasibility and cost assessment compared to total energy output.

Regardless of the technological underpinnings, there was a collective recognition that the transition to renewable energy is supposed to be a multidisciplinary effort that requires input from a variety of fields and stakeholders. The discussions highlighted the importance of co-creation and multidisciplinary collaboration to drive innovation in the renewable energy sector. None of the Landscape Architects felt that they alone would, could, or should be responsible for the entire process of PBAS energy development. Successful renewable energy projects require the active involvement of stakeholders from diverse backgrounds, including engineers, architects, energy companies, policymakers, environmental scientists, and community members. Participants recognized that further engagement with researchers, manufacturers, and education professionals on energy planning and technology requirements—along with increased transparency on associated data—will maximize community value. Participants recognized that when designers, engineers, and the community collectively reach an understanding, forward progress is guaranteed. Participants acknowledged that there is still a significant amount of work to be done to achieve a collective understanding in the realm of PBAS energy landscapes. However, the conference frequently pointed to how landscape architects, engineers, policy makers, and the community have successfully come together to make substantial progress on decentralized stormwater interventions in the Southwest.

## 4. Discussion

### 4.1. Novel Directions Introduced

The conference often pointed to the parallels and connections between renewable energy and water conservation. Currently, Landscape Architects have positioned themselves with strengths in stormwater management and water conservation and can re-tool their skillset and apply these efforts to energy systems. Participants suggested that this could involve measures such as avoiding undisturbed lands for hydrology, erosion, and aquifers, reducing evaporation through agrivoltaics, infrastructure over canals, and floatovoltaics at key reservoirs, and utilizing micro-hydro on existing water infrastructure. At a larger scale, the future of energy networks can parallel civic watershed networks and sub-watersheds by incorporating a balance between top-down and bottom-up influences.

In stormwater management, Landscape architects already play a pivotal role by balancing their expertise in site planning, urban design, and ecological considerations with decentralized, low-cost, high-return bioswale interventions. They have designed stormwater landscapes with features such as permeable pavements, green roofs, and rain gardens, effectively mitigating runoff and enhancing on-site infiltration. Collaborating with urban planners, landscape architects have helped integrate stormwater management seamlessly into communities, considering natural landscapes, water flow patterns, and potential flooding areas. The incorporation of biodiversity and ecological design further distinguishes landscape architects' approach to creating sustainable solutions that not only manage stormwater but also contribute to environmental balance. This environmental and ecosystem approach has been a natural niche for landscape architects to provide valuable contributions due to both their vocational and educational background as experts in plant material, ecology, and ecosystem restoration.

To play a similar role in the energy transition, landscape architects would need to extend their focus to integrating renewable energy infrastructure into landscape-based interventions. This could involve designing spaces for solar panels, wind turbines, and

other energy-generating technologies. Collaboration with urban planners, advocacy for sustainable practices, and a commitment to aesthetically pleasing and functional designs will be essential in driving the integration of renewable energy solutions into the everyday landscape of city and community life. The adaptation of landscape architects' skills in site planning, biodiversity considerations, restoration work, and policy advocacy can collectively contribute to a landscape-led approach to facilitating a successful energy transition. If these measures can be implemented at both an educational and professional level, it is possible for landscape architects to take a stronger role in shaping the future of green energy development as they work with other design professionals to design interventions that are wildly popular amongst the community where they are installed.

Participants recognized the significance of solar energy in the region's energy transition, despite concerns about its aesthetics. The potential for vertical opportunities and scaling in solar installations makes it an attractive option for economic development. However, the conference also highlighted the importance of considering alternative forms of renewable energy and prioritizing basic solutions in addition to new technologies. The value of shade and street trees, for instance, should not be underestimated, and policymakers need to carefully assess the long-term environmental consequences of renewable energy projects, such as erosion, impacts on hydrology, and effects on local species.

The conference emphasized the need for obtaining accurate metrics and data to accurately convey the story of renewable energy. Greenwashing was cautioned against, and participants stressed the importance of obtaining factual and comprehensive lifecycle analysis data from companies involved in green interventions. Landscape architects must be equipped with reliable and complete information to make informed decisions and create impactful energy landscapes. By doing so, they can contribute to the successful implementation of renewable energy projects and ensure a project truly has long-term sustainability and benefits.

The conference provided valuable insights and recommendations for energy landscape design in the southwestern US; however, it is important to acknowledge the limitations of the findings. As all the participants were from the southwestern region, the applicability of these solutions and ideas to other regions may require additional research, funding, and co-creation workshops. Energy landscapes elsewhere necessitate careful consideration of local contexts, resources, and community needs.

### 4.1.1. Siting

Landscape architects play a crucial role in the selection and siting of green energy interventions due to their extensive knowledge of site analysis [31,43]. Recognizing the significance of avoidance siting, in addition to proper siting, is fundamental to adopting a PBAS approach for renewable energy development [31]. By strategically linking renewable energy development with energy and landscape conservation, landscape architects can contribute to the long-term sustainability of renewable resources.

In pursuing a PBAS approach, landscape architects prioritize passive landscape solutions that go beyond energy production. Picking landscapes that can serve as energy-generation landscapes while remaining desirable places for the community to visit and interact with is a key part of multi-functional siting [32]. By understanding the interconnectedness of various components within a community or place, landscape architects can propose interventions that balance energy consumption with other community needs. This comprehensive approach not only contributes to green energy generation but also fosters a sense of peace and harmony among community members, avoiding potential conflicts that may arise from less considerate energy development.

The expertise of landscape architects in site analysis enables them to identify suitable locations for green energy interventions that align with the unique characteristics of a specific site [43]. By considering factors such as topography, microclimates, natural resources, and cultural values, landscape architects can propose renewable energy solutions that seamlessly integrate into the existing landscape fabric. This holistic approach ensures

that renewable energy development is contextually appropriate and harmonious with the surrounding environment, maximizing the benefits while minimizing any potential negative impacts [53].

In conclusion, landscape architects, with their deep understanding of site analysis and expertise in holistic design, are uniquely positioned to contribute to the selection and siting of green energy interventions. By recognizing the significance of avoidance siting and prioritizing passive landscape solutions, landscape architects can advance the goals of a PBAS approach for renewable energy development, accounting for the needs of both the community and the environment [31].

### 4.1.2. Diversifying

The deployment of a diverse array of green energy technologies at a finer scale is crucial for achieving multiple objectives, including diversifying energy portfolios, enhancing aesthetics and well-being, and capitalizing on the robustness of place in cities, towns, and communities. Landscape architects, with their expertise in site analysis and selection, are uniquely positioned to match the appropriate green energy interventions to specific sites, similar to how they select the right plants for the right site [36,43]. This parallel between plant selection and energy intervention highlights the importance of considering site characteristics and context when deploying renewable energy technologies.

In contrast to the monocultural deployment of solar photovoltaics, which can lead to a downgrading of landscape diversity in desert areas, landscape architects can contribute to the integration of a variety of solar configurations that respect and enhance the urban, suburban, and peri-urban characters of different places [31,54]. By carefully analyzing the unique qualities and context of each site, landscape architects can select the most suitable solar configurations that complement the existing built environment and landscape, thereby ensuring a harmonious integration of green energy technologies.

Moreover, landscape architects can help in identifying and utilizing renewable energy technologies in already disturbed areas of a region. Rather than encroaching on naturalistic landscapes, these disturbed areas, such as brownfields or industrial sites, present viable options for the integration of renewable energy interventions through the process of co-location [55–57]. Landscape architects can help navigate the complexities of such sites, considering factors such as land availability, access to infrastructure, and potential environmental remediation requirements. These skills help to determine the most appropriate renewable energy technologies that align with the specific site conditions and support the overall sustainability goals.

The remediation of brownfields and derelict energy generation sites is a crucial step in promoting energy justice. Landscape architects have a track record of successfully transforming brownfields into vibrant environments using bioremediation, often repurposing once-derelict sites into public parks. Additionally, some parks even include the remains of legacy energy generation facilities, serving as a direct reference to the history of the site. A compelling illustration of this approach is Gas Works Park in Seattle, Washington, designed by landscape architect Richard Haag [58]. Through an innovative and ecological approach, Haag transformed a former coal plant site into a public park, seamlessly blending industrial remnants with green space. Gas Works Park, opened to the public in 1975, has since become a popular and iconic landmark in Seattle [58]. This example underscores the transformative capacity of landscape architects to turn forgotten energy production wastelands into cherished and critically acclaimed sites. Gas Works Park not only rectified environmental damage but also provided a stunning public space, offering a picturesque retreat by the water. Importantly, it stands as a testament to the regenerative impact of landscape architecture on neglected areas, benefiting the entire community.

This specific instance showcases landscape architects' capacity to remediate past damage and highlights the valuable role landscape architects can play in addressing historical injustices associated with conventional energy development. Moreover, by remediating a previously compromised site, landscape architects can help contribute to

preserving and safeguarding untouched land by preventing it from being developed for new energy landscapes. Essentially, landscape architects can address three challenges simultaneously: mitigating a legacy brownfield, creating a community-oriented energy landscape, and protecting wild regions from unnecessary development.

Landscape architects' expertise in site analysis and selection positions them as valuable contributors to the deployment of green energy technologies, as they can merge functional engineering standards while simultaneously optimizing ecological benefits and fostering additional recreational opportunities [59]. Just as they match plants to their ideal environments, landscape architects can carefully assess the characteristics and context of sites to identify and deploy a diverse array of renewable energy technologies.

### 4.1.3. Local Identity

Maintaining the local identity of communities is a crucial consideration when implementing renewable energy development. Landscape architects have a unique opportunity to invest the time and effort to understand the communities in which green energy interventions take place. By engaging with the local context, landscape architects can design spaces and places that not only fulfill energy needs but also enhance the overall aesthetics and well-being of the community. Community members often turn to aesthetics when prompted to comment on their emotions and considerations for public green energy system-based interventions [60].

Aesthetics as a category does play an important aspect in fostering community acceptance and support, as well as ensuring that renewable energy development is viewed as an opportunity for improvement rather than degradation by the members of the region where an intervention takes place [61]. The phrase NIMBY has long been in the public lexicon for signifying "Not In My Backyard", and NIMBY can cause many project terminations [62,63]. Research has noted that NIMBY can act as a force for social good as it can prompt developers and energy professionals to create cleaner, more environmentally friendly, and more aesthetically acceptable structures to garner community support [62,63].

One key strategy that conference participants suggested to avoid NIMBY was to maintain local identity. Participants suggested that maintaining local identity can involve incorporating landscape features that align with the community's values and aspirations. Landscape architects can integrate water harvesting systems, native plant materials, and other fitting landscape improvements into the design of green energy PBAS interventions [64]. By considering the specific characteristics and needs of each community, these additional landscape features can contribute to a socially acceptable approval process and help improve degraded or neglected sites.

By designing renewable energy projects that are not only functional but also visually appealing, landscape architects can create spaces that the community can genuinely appreciate and embrace [61]. Projects that cater to community concerns will have a higher chance of gaining community acceptance and support, as they demonstrate a commitment to design quality and a consideration for the overall environment, culture, and aesthetics of the community [61]. This approach can help alleviate potential resistance and resentment that may arise from the introduction of new energy interventions. Instead, landscape architects can create designs that are desired by the community, making the renewable energy development process more collaborative and inclusive.

Landscape architects play a vital role in maintaining local identity during renewable energy development. By spending time to understand the communities and engaging with the local context, landscape architects can design green energy interventions that not only meet energy objectives but also enhance the aesthetics and well-being of cities, towns, and communities. This collaborative PBAS approach is crucial for creating a sustainable and socially acceptable transition to a greener energy future.

4.1.4. Beyond Energy Optimization

Landscape architects have a unique role in ensuring that these interventions not only function as energy generation sites but also contribute to the creation of interesting and effective landscapes. While energy optimization is essential, it is equally crucial to consider other landscape performance metrics that bring value to the community and the overall aesthetic quality of the environment. Landscape architects can play a vital role in finding the right balance between energy efficiency and creating memorable, desirable community spaces. This approach recognizes that community buy-in is the most significant factor in the successful rollout of green energy initiatives.

When designing renewable energy projects, landscape architects should consider how orientation and angles can harmonize with the site and contextual forms to achieve aesthetic performance. By integrating these considerations into the design process, renewable energy interventions can become visually appealing elements within the landscape. This focus on aesthetics has soft benefits that go beyond visual appeal; it can help alleviate concerns and resistance from the community. By creating landscapes that are both functional and visually pleasing, landscape architects can foster a sense of pride and ownership among community members, leading to greater acceptance and support for renewable energy developments [32].

To ensure community buy-in and acceptance, landscape architects can play a crucial role in educating the public about the benefits and trade-offs associated with renewable energy projects. By fostering a better understanding of the environmental, social, and economic advantages of these interventions, landscape architects can generate local innovation and engagement. By involving community members in the design process and providing them with the knowledge to make informed decisions, landscape architects can encourage the development of more robust and contextually appropriate renewable energy solutions. This approach not only empowers communities but also helps build trust and cooperation between stakeholders.

Landscape architects at the conference expressed an ability to craft renewable energy developments as both energy generation sites and interesting landscapes. By considering landscape performance metrics beyond energy optimization, such as aesthetics and community value, landscape architects can create memorable and desirable community spaces [31,65]. While energy efficiency is important, multiple participants expressed a willingness to sacrifice some efficiency to ensure that renewable energy projects are embraced by the community. The participant's logic was positioned behind the realization that an energy landscape is not effective if it cannot be built due to community resistance. By harmonizing energy performance with aesthetics, educating the public, and fostering local innovation, landscape architects can play a pivotal role in securing community buy-in for green energy initiatives. This collaborative approach is essential for creating sustainable and successful renewable energy developments that meet the needs and aspirations of both communities and the environment.

The powers involved in shaping a new energy typology are diverse and span across community engagement, design-driven planning, and government regulations. While landscape architects can hold influential design and creative powers in envisioning place-based renewable energy solutions, their legal authority to actively step into a central role depends on the jurisdiction and specific regulations in place. A landscape architect's ability to guide energy transitions will be influenced by existing frameworks and policies. The government will always play a pivotal role in shaping the legal and regulatory landscape for clean energy initiatives. Its powers extend to creating frameworks that encourage the use of PBAS design methodologies and the inclusion of core principles like meaningful multifunctionality, respect for natural capital, local value generation, decentralization of energy generation, resilience against climate disruptions, and the promotion of energy justice. Effective collaboration between landscape architects, architects, engineers, communities, and government entities is essential for navigating this complex terrain and ensuring a

harmonious transition to a clean energy future that resonates with the values and unique characteristics of diverse communities.

## 5. Conclusions

We began this research with a question about the prospective contributions of landscape architects to the development of renewable energy infrastructure. To underpin this inquiry, a literature review was conducted, encompassing historical energy trends and emerging methodologies. Subsequently, a student ideas competition was organized, focusing on the identified PBAS energy methodologies from the literature review. Then, a Southwest Energy Conference was hosted to facilitate collaborative ideas, strategies, and principles. The conference involved a diverse assembly of design and energy professionals, all converging around the aforementioned PBAS energy methodologies. Finally, a brief analysis of participants' contributions was performed, aligning them with insights derived from the literature review.

Consequently, a distinctive collection of co-created ideas for future exploration has been formulated, providing deeper insights into the potential role that landscape architects may play. The conclusions drawn from our literature review and the co-creation process during the SRVW have provided valuable insights into the design process of renewable energy development in the Southwest. The SRVW highlighted the success of the Green Stormwater Infrastructure movement as an example of how landscape architects can add value, engage communities, and change public perceptions of renewable and climate-sensitive projects [59,65]. With their understanding of ideal site locations and the ability to co-create spaces with the community, landscape architects can ensure that new energy interventions align with the six pathways and deliver benefits for all stakeholders.

As we strive towards a more sustainable future, landscape architects have a unique opportunity to expand their expertise in the field of renewable PBAS energy landscapes and participate in a sector-wide rollout of green energy sites. By enhancing curricula, fostering familiarity with energy systems, and pushing for professional growth, landscape architects can help design cost effective landscape-based energy interventions that maintain community support and increase perceived infrastructure quality [66].

The conference findings indicate that Landscape architecture also offers a framework to increase community acceptance of energy landscapes by transforming renewable energy projects into engaging, unique, and productive infrastructure. By integrating various considerations from a generalist landscape perspective, landscape architects can ensure that these projects connect with their surroundings and provide multiple benefits beyond energy generation [59].

Central to the landscape architecture approach towards PBAS energy landscapes is the focus on community and place. Landscape architects understand the importance of meaningful community engagement and the need to align renewable energy projects with the values and aspirations of local residents [65]. Through conceptualizations and graphic presentations, landscape architects create a platform for discussions among consultants and community members. This participatory process fosters dialogue, evaluation, and the exploration of acceptable compromises and consensus. By involving the community in the design process, landscape architects ensure that the energy landscape becomes a reflection of the local context and a source of pride for the community [32,65].

The conference has found that there are many ways for Landscape architects to help shape renewable energy projects as engaging and productive infrastructure. Their expertise in community engagement, PBAS design, and interdisciplinary collaboration positions them as catalysts for designing energy landscapes that go beyond mere functionality [31]. By viewing renewable energy projects through a landscape lens, landscape architects can ensure that these projects integrate seamlessly with their surroundings, provide multiple benefits, and garner community support. This integrated approach can contribute to the successful implementation of renewable energy initiatives and foster a more sustainable and aesthetically pleasing built environment. At the conference, participants suggested that

landscape architects and design professionals should seek greater alignment of their skills with green energy and focus on the capacity to impact the world within their sphere of influence, as well as take an active approach to the public sector and try to influence policies in a positive direction towards increased utilization of renewable energy. The conference discussions also highlighted the importance of multidisciplinary collaboration and the need for alignment and collaboration to achieve a sustainable energy future in the Southwest.

In conclusion, landscape architects have an incredible opportunity to contribute to a renewable energy future by designing and implementing PBAS interventions that transform the management of civic resources at the community level. Building on their expertise in stormwater management, ecology, and community engagement, landscape architects can expand their knowledge around energy. By embracing this transformative opportunity and collaborating with communities, landscape architects have the potential to aid in the creation of PBAS energy landscapes, driving society towards a more sustainable and resilient energy future.

**Author Contributions:** Conceptualization, K.D.; methodology, K.D., W.G. and K.P.; software, K.P.; validation, W.G., K.D. and K.P.; formal analysis, W.G.; investigation, W.G., K.D. and K.P.; resources, K.D.; data curation, K.D.; writing—original draft preparation, W.G. and K.P.; writing—review and editing, W.G.; visualization, K.D.; supervision, K.D.; project administration, K.D.; funding acquisition, K.D. All authors have read and agreed to the published version of the manuscript.

**Funding:** This research was funded by US Department of Energy's Office of Energy Efficiency and Renewable Energy (EERE) Water Power Technologies Office (Contract DE-AC05-76RL01830). The views expressed herein do not necessarily represent the views of the US Department of Energy or the United States Government.

**Institutional Review Board Statement:** Not applicable.

**Informed Consent Statement:** Not applicable.

**Data Availability Statement:** No new data were created or analyzed in this study. Data sharing is not applicable to this article.

**Acknowledgments:** The authors wish to acknowledge collaborative contributions for the workshop from Danielle Preziuso, Simon Gore, Kenneth Kokroko, Jonathan Bean, Adriana Zuniga-Teran, a, and Omar Youssef, as well as the valuable contributions from the workshop attendees.

**Conflicts of Interest:** The authors declare no conflicts of interests. The funders had no role in the design of the study; in the collection, analyses, or interpretation of data; in the writing of the manuscript; or in the decision to publish the results.

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
