# Peer review of "The Power of Place: Unleashing the Potential of Place-Based Green Energy Landscapes"

_2673-8945, doi:10.3390/architecture4010010_

Round 1

Reviewer 1 Report

Comments and Suggestions for Authors

The manuscript presents the results of a workshop that centers around the potential role of landscape architects in the design of renewable energy landscapes in the Southwest United States. The conclusions are drawn based upon a literature review and the workshop. The manuscript is rather argumentative (no extensive discussion of the findings in a larger scientific context) and a large part of the results is dedicated to the findings of the workshop.

The manuscript fits well within the aim and scope of the journal Architecture as it relates to both academic research and design practice. As mentioned before, the manuscript is argumentative in a sense that it strongly argues for a pivotal position of landscape architects in the transition to a renewable energy system. This also seems to fit the journal rather well. The topic is very relevant as well as the arguments the authors aim to make in this manuscript. However, there are some issues that should be addressed to present the research more transparent and evidence-based. Below, I’ve listed some major remarks, followed by a set of more remarks related to specific parts of the manuscript. I wish to encourage the authors to continue on this topic by revising this manuscript and make it interesting for a larger international audience.

Major remarks

-      The entire manuscript is written with the scope on the Southwest United States. This is perfectly fine, but to readers from outside this geographic region, two things are lacking: (1) a proper description of the (geographic, political, environmental, …) background of this region. (2) reflection of the findings in this region to what is already known and happening in other parts of the USA and the world.

-      Related to the point above: what is currently the role of landscape architects in the context of Southwest United States? The manuscript mentions their active role in stormwater management, but what is their specific role in this topic? And what is the current (and past) role of landscape architects in the development of fossil energy landscapes?

-      The manuscript uses several key concepts, mainly ‘place’ (place – based), ‘scale’ (at scale or appropriately scaled), landscape and (the role of ) landscape architecture. Unfortunately, none of these concepts are properly explained or defined. This leaves too much room for interpretation and leads to relatively unspecific conclusions.  

-      Very broad themes are addressed in a single-day workshop. Also, for some of these themes one could argue easily the landscape architect is not the discipline to be in the lead (e.g. energy justice, or decentralization). The discipline could for sure play a (large) role in these themes, but other disciplines seem more likely to be in the lead (e.g. governance experts, process managers,  social experts). If the authors do like to claim this point, please be more specific in why this would be the case.

-      The Methods and Materials section lacks clarity on the workshop. For example, how many participants were present? Which disciplines were represented and in which numbers? Was the workshop fully on-site, or also partially online (there are some references to a ‘virtual workshop’)?

-      Related to the point above: the workshop aimed to ‘unite landscape architects and energy professionals’, but in the outreach, it seems mostly landscape architects were invited? If for a large majority only landscape architects were present at this workshop, this greatly influences how the results should be read. To be a bit frank and direct: is there a lot of wishful thinking at a workshop full of landscape architects all arguing their profession should be in the lead in this topic? Furthermore, the authors do not explain how data was collected during the workshop and what kind of analysis was performed on the collected data.

-      There is little reference to a growing body of (international) literature on this topic. The discussion is very short and contains hardly references, and also the introduction is very limited with regard to recent research on the topic of energy landscape design. Quite some landscape architecture research groups (e.g. in Europe and beyond) have already studied the topic of energy landscapes for over a decade.

Specific remarks

-      The introduction contains quite some explanations of the rationale of the workshop, but uses (slightly) different terms and explanations. For example: workshop (line 43), regional workshop (line 45), Southwest Regional Virtual Workshop (line 54), conference (lines 122 and 124). I assume these all refer to the same event? If so, please in the first instance mention the name in full and use a short version consistently in the remainder of the text. Another example, there are multiple aims/objectives mentioned for the workshop, some more explicit than others, but all not the same: ‘the regional workshop serves to help foster the co-creation of strategies and principles … to help move forwards fostering a new energy perspective for our region and nation (line 45-48) vs. ‘the objective was to seek stakeholder expertise …’ (line 51-52) vs. ‘the purpose of … is to connect landscape architects and related design professionals’ (line 54-55) vs. ‘the workshop’s mission is to provide an environment for participants to co-create …’ (line 57 onwards). I strongly advise to revise this section and clearly explain the objective of the workshop once and then go into more detail.

-      The described problem statement (line 70-72) is too vague considering the current knowledge state of the topic of this manuscript. The authors now mention scholars agree on the ‘why’ but there is less certainty on the ‘how’ to transition towards renewable energy. This is very broad and must be much more tailored to the specific topic of this manuscript.

-      The authors suggest a ‘literature review’ has been done (line 73) that has helped to create ‘a framework suggesting that Landscape architects could be a key contributor to balancing community and commercial desires in the renewable energy sector through effective design and planning’. The role and position of this ‘literature review’ is ambiguous and thus not clear to the reader. In my opinion, either the literature review is substantial (up to a systematic review) and is part of the used methods, or the literature review is the base from which the authors introduce the topic and argue something is not yet studied. Considering the methods and materials section, I believe the former is implied in this manuscript. However, the authors claim a ‘framework’ has been developed based upon this review, which would imply this framework is part of the results (which it is not). Please address these issues.

-      The acronym PBAS is introduced in line 133, but only explained in line 204. Please make sure the acronym is explained at the first mention. Do the same for acronyms like ASLA, CELA, NAMLA. These may be known to some part of your target audience, but not all.

-      Where do the ‘key questions’ mentioned in lines 135-141 come from? How are they linked to the six design principles mentioned earlier?

-      Lines 164-165: why is this virtual introduction activity mentioned? What is the relevance for this manuscript.

-      Lines 166-173: what was the design brief for the students assignment? On what criteria were the submissions evaluated? Please be more specific here, because that will help make more clear what the role (and impact) of the student work was in the workshop.

-      Line 353-356: the comparison with the role of landscape architects on the topic of stormwater management is made multiple times. A solid comparison would help a lot to be more specific about the role LA plays in this topic and what would be needed to attain a similar role in the energy transition.

-      Line 444-448: isn’t this point in conflict with the theme of ‘energy justice’? Distributional injustice is a large topic in the energy transition, but here it seems landscape architects can play a role in strengthening this injustice … (over burdening certain places compared to others).

-      Line 474-480: this section (and others) could highly benefit from a built example in the USA or elsewhere to make this point more viable.

-      Section 3.4.4: this section should include some reflection on the governance of ‘beyond energy optimization’. How does this come into being? What are the powers involved? Does the landscape architect have the legal power to step into this role? What is the role of the government?

-      Section 3.4.5: this section is titled ‘renewable energy as landscape’, but large parts of this section seem to be more of a summary or conclusion to the entire paper, and hardly touches upon the concept ‘landscape’.

Comments on the Quality of English Language

See above.

Author Response

Thank you for the feedback; the author believes it has significantly enhanced the quality and relevance of the manuscript.

Reviewer 2 Report

Comments and Suggestions for Authors

The contribution addresses a very current and widespread topic in the scientific debate. It identifies coherent literary references and proposes an investigation methodology based on the specific tools of the architectural project (conferences and workshops).

It carries out a careful examination of the outcomes of the collegial activities, deducing from these the reference principles for the definition of the future guidelines of the project in the area of intervention under study.

It also highlights the need to involve additional scientific, institutional, territorial and industrial partners in the design and decision-making process, in order to increase the feasibility and social acceptance of project proposals.

In the final discussion, however, it does not propose a revision of the design and decision-making process analyzed and applied, in order to make it consistent with the considerations that emerged in the previous reflections.

 The authors underline the role of the landscape architect, also widely described in the previous paragraphs.

It would be useful to be able to verify the theoretical assumptions through the graphic exemplification of the design proposals that emerged during the workshop.

Author Response

Thank you for your comments! We have improved our citations and more clearly presented our conclusions.

  1. In the final discussion, however, it does not propose a revision of the design and decision-making process analyzed and applied, in order to make it consistent with the considerations that emerged in the previous reflections.
    • In the concluding paragraph, the authors now revisit the study objectives and the decision-making process. They explicitly articulate the insights gained through reflection on the preceding segments of the research.
  2.  The authors underline the role of the landscape architect, also widely described in the previous paragraphs.
    • In the conclusion, there is a section on the role of the landscape architect, emphasizing how the preceding steps have provided valuable insights into defining and understanding the "role of the landscape architect."
  3. It would be useful to be able to verify the theoretical assumptions through the graphic exemplification of the design proposals that emerged during the workshop.
    • The authors have integrated various graphics to emphasize the insights gained during the workshop. While they have refrained from featuring detailed images of the Concept Board drawing due to potential copyright concerns with shared images and links, they have chosen to present the research and process through a set of condensed graphics that encapsulate the covered topics.

Reviewer 3 Report

Comments and Suggestions for Authors

The authors of the article addressed the issue of shaping renewable energy infrastructure in harmony with the natural landscape with the support of specialist knowledge and skills of landscape architects. The issue is very current, and at the same time requires exploring and considering various conditions in order to ensure spatial order and aesthetics of environments, both when implementing innovative energy solutions and modernizing existing ones. It is also important for the well-functioning and acceptance of societies.

The authors analyzed the problem based on previous research (presented in Chapter 1) and workshops indicating certain ideas, paths and design strategies (presented in Chapter 2). Chapter 3 includes "Results" divided into subchapters.

Comments and suggestions:

Chapter 1 should be enriched with additional literature sources, especially since the authors emphasize the interdisciplinarity of the issue.

It is worth supplementing Chapter 2 with a graphic diagram illustrating the structure, course and action plan of the workshops.

Chapter 3 should be titled "Results and Discussion".

Point 3.4. "Novel Directions Introduced" together with sub-points can be treated as a separate chapter, which contains clear design guidelines, helpful for specialists in various design areas (urban planners, architects, landscape architects, constructors, designers of electrical, energy, water, heat, ventilation, air conditioning installations, etc.). In my opinion, this part treated the topic too generally, without providing specific, precise and structured design recommendations. Supplementation in the form of graphics, illustrations and photos would also be desirable. Illustrated examples and proposals for design solutions (even in the form of ideas, concepts or diagrams) would increase the scientific and practical value of the article. Chapter 4 "Discussion" might be better titled "Summary" or "Conclusions".

Author Response

Thank you for your feedback!

  1. Chapter 1 should be enriched with additional literature sources, especially since the authors emphasize the interdisciplinarity of the issue.
    • The author acknowledges the need for an expanded literature review and has accordingly enriched the manuscript with a more comprehensive examination of the issue, considering perspectives from politics, the public, energy officials, and further incorporating the viewpoints of design professionals. The literature cited has increased over 4 fold.
  2. It is worth supplementing Chapter 2 with a graphic diagram illustrating the structure, course and action plan of the workshops.
    •  The author agrees, and the report now incorporates a graphic diagram illustrating the structure of the event, overlaid with an action plan for the workshop.

  3. Chapter 3 should be titled "Results and Discussion".
    •  The author agrees and the change has been made.
  4. Point 3.4. "Novel Directions Introduced" together with sub-points can be treated as a separate chapter, which contains clear design guidelines, helpful for specialists in various design areas (urban planners, architects, landscape architects, constructors, designers of electrical, energy, water, heat, ventilation, air conditioning installations, etc.). In my opinion, this part treated the topic too generally, without providing specific, precise and structured design recommendations. Supplementation in the form of graphics, illustrations and photos would also be desirable. Illustrated examples and proposals for design solutions (even in the form of ideas, concepts or diagrams) would increase the scientific and practical value of the article.
    • The authors have implemented the suggested structural and organizational changes in addition to incorporating extra graphics into the document. It is essential to emphasize that the methodology employed during the single-day virtual conference facilitated the generation of general proposals, preliminary directions for future research, and rudimentary diagrams.  This limitation is duly recognized.
    • The authors have included new graphics that contain an Eisenhower matrix of specific directions suggested by participants.
    • Authors can include closer images of the Concept Board drawing but have held off as much of the images and links shared may be subject to copywrite issues if prominently displayed. For now authors have opted to showcase the research and process is a series of distilled graphics based upon the topics covered.

Reviewer 4 Report

Comments and Suggestions for Authors

The topics discussed in the article are very interesting and useful to the scientific community for a new approach to sustainability. However, the article is too general and does not show the concrete results of the workshop in terms of landscape. Therefore, it is requested to detail the article by showing the concrete results of the workshop as an accompaniment to the co-creation process. It would be very helpful to verify the design ideas that came up by illustrating them with imaginils, drawings, etc.

Author Response

Thank you for your feedback.

  1. The topics discussed in the article are very interesting and useful to the scientific community for a new approach to sustainability. However, the article is too general and does not show the concrete results of the workshop in terms of landscape. Therefore, it is requested to detail the article by showing the concrete results of the workshop as an accompaniment to the co-creation process. It would be very helpful to verify the design ideas that came up by illustrating them with imaginils, drawings, etc.
    • The authors have expanded upon the conferences findings through the inclusion of several new graphics and participant work showcasing some of the ideas generated.

    • Authors can incorporate more detailed images of the Concept Board drawings. However, they have refrained from doing so, as many of the images and links shared may be subject to copyright issues if prominently displayed. At present, authors have chosen to present the specific research findings through separate overview graphics. They are open to expanding on this by creating new graphics or featuring targeted close-ups of participant work after verifying the absence of copyright issues.

Round 2

Reviewer 1 Report

Comments and Suggestions for Authors

In general, the authors have addressed the main remarks following the original submission. The results are now stronger linked to the specific context of the followed method (the workshop). This gives the reader a more honest view on where the quite advocative conclusions are based upon. In some cases, the authors may have been a bit too thorough and overcomplete in addressing the comments, leading to some lengthy additions.

I would advise the authors to consider the following comments:

-        PBAS (place based at scale). Although the acronym/concept is now explicitly explained, the entire concepts remains a bit ambiguous, specifically in the context of the field of landscape architecture. Drawing from local (place) identities is one of the axioma’s in landscape architecture, but in this manuscript it seems this is specifically linked to the ‘technology’, not to the energy landscape as a whole (technology + landscape integrated). The example of the phone case is illustrative in this regard, because it seems to point to an approach in where the outer shell of technology is adapted to resemble something of local identity. E.g. yellow/brownish wind turbines in desert like landscapes vs. green turbines in forested areas. This example is not mentioned, but this is the idea I get when this concept is introduced. The acronym leads me to think this is some kind of gimmick/quick-fix, because (unless I missed something) there is not a more specific example of a PBAS energy landscape mentioned in the manuscript. Can the authors make this more concrete? E.g. refer to build projects that are in line with this PBAS approach? Or identify (if and) how the student designs adhere to this approach?

-        The extension of the introduction (line 25-141) is welcomed, but should be shorter to bring the reader quicker to the core of the manuscript.

-        Unfortunately, the numbering of the references were incorrect for this version, leaving it impossible to see what new references was used in what context.

-        Line 130-136: this sentence is far too long.

-        Line 137-141: why is the new funding allowing for interdisciplinary work? Has the budget been specifically labelled for this kind of work? Proper funding in itself is unfortunately not a guarantee for cross-sectoral thinking.

-        Line 295-309: “Submissions were … end of November”. I appreciate the more detailed description of the student work, but this part is too detailed and does not need to be in the manuscript.

-        Line 306: Typo (SSRVW) > there are some more throughout the manuscript, please have a proper check.

-        Figure 3 in this form is not needed for me, because the content is not readable.

Comments on the Quality of English Language

Fine, some typos (see overall comments).

Author Response

The authors appreciate your feedback and have made the suggested changes!

See Below:
_________________________________________________________________

In general, the authors have addressed the main remarks following the original submission. The results are now stronger linked to the specific context of the followed method (the workshop). This gives the reader a more honest view on where the quite advocative conclusions are based upon. In some cases, the authors may have been a bit too thorough and overcomplete in addressing the comments, leading to some lengthy additions.

==

The authors have trimmed content from each introduction section and made efforts to reduce wordiness throughout the article.

_________________________________________________________

The extension of the introduction (line 25-141) is welcomed, but should be shorter to bring the reader quicker to the core of the manuscript.

==

The authors have condensed the section to improve the pacing without sacrificing the necessary political, environmental, geographic, historical, and social introduction to the region, energy landscapes, and PBAS design philosophy.

_________________________________________________________

PBAS (place based at scale). Although the acronym/concept is now explicitly explained, the entire concepts remains a bit ambiguous, specifically in the context of the field of landscape architecture. Drawing from local (place) identities is one of the axioma’s in landscape architecture, but in this manuscript it seems this is specifically linked to the ‘technology’, not to the energy landscape as a whole (technology + landscape integrated). The example of the phone case is illustrative in this regard, because it seems to point to an approach in where the outer shell of technology is adapted to resemble something of local identity. E.g. yellow/brownish wind turbines in desert like landscapes vs. green turbines in forested areas. This example is not mentioned, but this is the idea I get when this concept is introduced. The acronym leads me to think this is some kind of gimmick/quick-fix, because (unless I missed something) there is not a more specific example of a PBAS energy landscape mentioned in the manuscript. Can the authors make this more concrete? E.g. refer to build projects that are in line with this PBAS approach? Or identify (if and) how the student designs adhere to this approach?

==

The authors have chosen a specific, concrete site that illustrates PBAS principles and added a new figure to help the reader visualize and comprehend our vision for PBAS sites. It is difficult to find many examples of PBAS landscapes, because the concept is new and has not been widely adopted or implemented in energy site projects. We hope that this conference findings and this report will promote the PBAS approach to energy landscape typologies. 

We appreciate the concerns you raised about the need for more specific examples to showcase what PBAS could mean. PBAS sites are not meant to imply a gimmick or a quick-fix for communities, and we have expanded the phone case analogy as well as added a landscape architecture example of how this concept has been well applied to better elucidate our point. Our priority in introducing this concept is to promote its potential for widespread adoption, customization, and user satisfaction on a massive scale, and we are hopeful that the additions have improved the manuscript.

__________________________________________________________________

Line 130-136: this sentence is far too long.

==

The authors agree and have revised this sentence.

__________________________________________________________________
 Line 137-141: why is the new funding allowing for interdisciplinary work? Has the budget been specifically labelled for this kind of work? Proper funding in itself is unfortunately not a guarantee for cross-sectoral thinking

 ==

 The authors recognize this concern and made revisions to the paragraph accordingly. They have removed the emphasis on funding as a driver of cross-sectional thinking and interdisciplinary work. Instead, the paragraph now takes a more nuanced approach, emphasizing public and private involvement without implying any mandate for cross-sectoral thinking.

 _________________________________________________________________

 Line 295-309: “Submissions were … end of November”. I appreciate the more detailed description of the student work, but this part is too detailed and does not need to be in the manuscript.

 ==

 The authors understand and have removed the section.

 _________________________________________________________________

 Line 306: Typo (SSRVW) > there are some more throughout the manuscript, please have a proper check.

 ==

 The authors have corrected this and looked through the document to fix other typos across the manuscript.

 _________________________________________________________________

 Figure 3 in this form is not needed for me, because the content is not readable.

 ==

 The authors removed Figure 3 and replaced it with two close-ups to illustrate an example of the discourse style observed at the conference.

 ________________________________________________________________

 Unfortunately, the numbering of the references were incorrect for this version, leaving it impossible to see what new references was used in what context.

 ==

 The authors thoroughly apologize and have corrected this error.

__________________________________________________________________

Reviewer 3 Report

Comments and Suggestions for Authors

I would like to thank the authors for correcting and supplementing the manuscript, taking into account my previous comments and suggestions.

I have the following comments about the current version:

1. Please consider adding the names of the students who created the drawings and renderings in the caption of Figure 1. In my opinion, the drawing (6) placed in the lower right corner differs graphically from the rest (it is too sketchy). Maybe it's worth maintaining harmony and aesthetics?

2. Figure 3 is illegible due to the small scale of individual elements, which means that it provides little information to the work. Please consider enlarging the illustration, dividing it into parts or simplifying it (e.g. by removing unnecessary details).

3. Please correct the numbering of references.

Author Response

The authors appreciate your insight, apologize for the citations issue, and have made all suggested changes. Please see below:

______________________________________________________________
Please consider adding the names of the students who created the drawings and renderings in the caption of Figure 1. In my opinion, the drawing (6) placed in the lower right corner differs graphically from the rest (it is too sketchy). Maybe it's worth maintaining harmony and aesthetics?

==

The authors have replaced Example 6 with a different one. Furthermore, all student work has been properly credited with their names within the Figure description
_____________________________________________________________

Figure 3 is illegible due to the small scale of individual elements, which means that it provides little information to the work. Please consider enlarging the illustration, dividing it into parts or simplifying it (e.g. by removing unnecessary details).

==

The authors removed the Figure and replaced it with two close-ups to illustrate an example of the discourse style observed at the conference.
____________________________________________________________

Please correct the numbering of references.

==

The authors thoroughly apologize and have corrected this error.
____________________________________________________________

Reviewer 4 Report

Comments and Suggestions for Authors

I think the authors have improved the presentation of the work enough. However, I very much regret that I cannot see, because they are copyrighted, more images of the designed landscapes.

Author Response

The authors note your feedback and have attempted to showcase more of the participants work from the conference in two new figures. 

_______________________________________________________

I think the authors have improved the presentation of the work enough. However, I very much regret that I cannot see, because they are copyrighted, more images of the designed landscapes.

==
The authors have included two close-ups to exemplify the discourse style observed at the conference and the ideas discussed. Since it was a brief idea co-creation conference, no finished digital renders or construction documentation for energy landscapes were produced. Our copyright concerns were around shared images that participants uploaded to concept board.
____________________________________________________________________________